# Branched Chain Amino Acid Supplementation to a Hypocaloric Diet Does Not Affect Resting Metabolic Rate but Increases Postprandial Fat Oxidation Response in Overweight and Obese Adults after Weight Loss Intervention

**DOI:** 10.3390/nu13124245

**Published:** 2021-11-26

**Authors:** Delicia Shu Qin Ooi, Jennifer Qiu Rong Ling, Fang Yi Ong, E Shyong Tai, Christiani Jeyakumar Henry, Melvin Khee Shing Leow, Eric Yin Hao Khoo, Chuen Seng Tan, Mary Foong Fong Chong, Chin Meng Khoo, Yung Seng Lee

**Affiliations:** 1Department of Paediatrics, Yong Loo Lin School of Medicine, National University of Singapore, Singapore 117549, Singapore; paeosqd@nus.edu.sg (D.S.Q.O.); qrongling@gmail.com (J.Q.R.L.); fangyi_o@yahoo.com (F.Y.O.); 2Khoo Teck Puat-National University Children’s Medical Institute, National University Health System, Singapore 119074, Singapore; 3Department of Medicine, Yong Loo Lin School of Medicine, National University of Singapore, Singapore 117549, Singapore; mdctes@nus.edu.sg (E.S.T.); mdckyhe@nus.edu.sg (E.Y.H.K.); mdckcm@nus.edu.sg (C.M.K.); 4Clinical Nutrition Research Center, Agency for Science, Technology and Research, Singapore 117599, Singapore; bchchjh@nus.edu.sg (C.J.H.); mary_chong@nus.edu.sg (M.F.F.C.); 5Singapore Institute for Clinical Sciences, Agency for Science, Technology and Research, Singapore 117609, Singapore; melvin_leow@sics.a-star.edu.sg; 6Department of Endocrinology, Division of Medicine, Tan Tock Seng Hospital, Singapore 308433, Singapore; 7Lee Kong Chian School of Medicine, Nanyang Technological University, Singapore 639798, Singapore; 8Saw Swee Hock School of Public Health, National University of Singapore, Singapore 117549, Singapore; ephtcs@nus.edu.sg

**Keywords:** branched chain amino acids (BCAA), resting metabolic rate (RMR), diet-induced thermogenesis (DIT), respiratory quotient (RQ), carbohydrate oxidation, lipid oxidation, hypocaloric diet, high-protein, weight loss, obese and overweight adults

## Abstract

Background: Branched chain amino acids (BCAA) supplementation is reported to aid in lean mass preservation, which may in turn minimize the reduction in resting metabolic rate (RMR) during weight loss. Our study aimed to examine the effect of BCAA supplementation to a hypocaloric diet on RMR and substrate utilization during a weight loss intervention. Methods: A total of 111 Chinese subjects comprising 55 males and 56 females aged 21 to 45 years old with BMI between 25 and 36 kg/m^2^ were randomized into three hypocaloric diet groups: (1) standard-protein (14%) with placebo (CT), (2) standard-protein with BCAA, and (3) high-protein (27%) with placebo. Indirect calorimetry was used to measure RMR, carbohydrate, and fat oxidation before and after 16 weeks of dietary intervention. Results: RMR was reduced from 1600 ± 270 kcal/day to 1500 ± 264 kcal/day (*p* < 0.0005) after weight loss, but no significant differences in the change of RMR, respiratory quotient, and percentage of fat and carbohydrate oxidation were observed among the three diet groups. Subjects with BCAA supplementation had an increased postprandial fat (*p* = 0.021) and decreased postprandial carbohydrate (*p* = 0.044) oxidation responses compared to the CT group after dietary intervention. Conclusions: BCAA-supplemented standard-protein diet did not significantly attenuate reduction of RMR compared to standard-protein and high-protein diets. However, the postprandial fat oxidation response increased after BCAA-supplemented weight loss intervention.

## 1. Introduction

Resting metabolic rate (RMR) and substrate utilization are crucial in maintaining energy balance, which underlies the regulation of body weight [1]. 

RMR, also termed as resting energy expenditure (REE), is defined as the energy required by the body when in resting condition without any physical activity [2]. RMR accounts for most of the body’s energy needs and expenditure [3], and differences in body composition influence an individual’s RMR [4]. Variability in the fat mass and lean mass (fat free mass) are associated with variability in RMR [5,6]. In particular, an increase in lean mass is correlated with a higher RMR [5,7]. 

It is well-established that energy expenditure, including RMR decreases after weight loss [8,9]. This reduction in RMR is known as metabolic adaptation (also termed adaptive thermogenesis) [10,11]. During weight loss, the energy expenditure exceeds energy intake resulting in a negative energy balance. Metabolic adaptation will be directed towards energy sparing, which is the lowering of RMR to compensate for the negative energy balance [12,13]. A low RMR is regarded as a risk factor for weight gain [14,15], and metabolic adaptation after weight loss is also a predictor of weight regain [16]. Hence, minimizing the loss of RMR during or after weight loss might reduce subsequent regain of weight. 

Substrate utilization is indicated by the respiratory quotient (RQ). A high fasting RQ corresponds to carbohydrate oxidation, while a low fasting RQ corresponds to fat oxidation [17]. Individuals with a high fasting RQ had greater gains in body weight and fat mass compared to a low/moderate fasting RQ among free-living healthy young adults [18]. RQ is positively correlated with total body fat mass, trunk fat mass, and visceral adipose tissues [19]. In addition, a lower postprandial RQ corresponding to higher fat oxidation is also associated with significant reductions in fat mass and body weight [20]. 

Protein is an essential macronutrient for building muscle mass (lean mass) [21], and a higher protein intake is associated with higher lean mass and muscle strength [22]. High-protein hypocaloric diets have been shown to preserve lean mass during weight loss [23,24,25,26], and a protein-enriched diet was found to mitigate the decrease in lean mass and RMR in obese subjects who underwent sleeve gastrectomy [27]. Branched chain amino acid (BCAA: valine, leucine, and isoleucine), which forms one-third of total muscle protein, can stimulate skeletal muscle regeneration [28] and suppress post-exercise muscle degradation [29,30]. In individuals who undergo resistance training, a BCAA-supplemented hypocaloric diet is able to maintain lean mass and preserve skeletal muscle performance while losing fat mass [31]. BCAA supplementation to a moderate protein hypocaloric diet was found to cause the highest body weight loss and decrease in percent body fat, as well as a significant reduction in abdominal visceral adipose tissue in a group of competitive wrestlers [32]. However, some studies did not observe an effect of BCAA supplementation on body weight, lean mass, and fat mass in overweight and obese adults after a hypocaloric diet-induced weight loss [33,34]. 

Preliminary studies showed that BCAA supplementation increased RMR in human adults [35,36]. The knockdown of branched-chain aminotransferase (BCAT) gene that encodes the enzyme involved in BCAA catabolism resulted in increased plasma BCAAs, decreased adiposity and body weight, and increased energy expenditure in mice [37]. In addition, dietary BCAA restriction has been found to alter substrate utilization in Zucker fatty rats [38], and BCAA supplementation was shown to enhance lipid oxidation during exercise [39]. However, the effect of BCAA supplementation on RMR and substrate utilization during hypocaloric diet-induced weight loss intervention in overweight and obese adults has not been investigated.

This study compared the change in RMR and substrate utilization between diet groups with different protein content supplemented with or without BCAA during weight loss. We hypothesize that a BCAA-supplemented standard-protein hypocaloric diet would minimize the decrease in RMR and alter substrate utilization during a weight loss intervention. 

## 2. Materials and Methods

### 2.1. Study Participants

The study participants were part of a randomized, controlled trial registered at clincialtrials.gov (accessed on 10 November 2021) as NCT02277275. The trial aimed to examine the effect of BCAA supplementation on changes in primary outcomes including lean and fat mass, and changes in secondary outcomes including insulin sensitivity, RMR, and diet-induced thermogenesis (DIT). We have previously reported the effect of BCAA supplementation on lean mass preservation and insulin sensitivity of the participants in the trial [34]. In this study, we included 111 subjects who had completed indirect calorimetry after 16 weeks of weight loss intervention. On the basis of mean change of RMR (−412 ± 67 kcal/day) after 8 weeks of calorically restricted diet supplemented with BCAA [31], we found our sample of 111 subjects to be more than adequately powered (>90%) at α = 0.05 to detect differences in RMR between hypocaloric diet groups supplemented with and without BCAA [40]. Out of these 111 subjects, 109 subjects completed the 24 week intervention study (Appendix A). The subjects were Chinese men and women, aged between 21 and 45 years old. They were either overweight or obese, with BMI ranging between 25 and 36 kg/m^2^, and with percent body fat ≥25%. All of them had no existing medical conditions (e.g., diabetes mellitus; hypertension; and kidney, liver, and thyroid problems). All procedures were in accordance with the Declaration of Helsinki, as revised in 2013. Written informed consent was obtained from all subjects, and this study was approved by the Domain Specific Review Board of National Healthcare Group, Singapore.

### 2.2. Study Design 

The study design for this trial has been described previously [34]. The study participants were block randomized by gender and BMI groups (25–29 kg/m^2^ versus 30–36 kg/m^2^) into three hypocaloric diet groups: (1) standard-protein diet with placebo supplementation (control, CT), (2) standard-protein diet with BCAA supplementation (BCAA), and (3) high-protein diet with placebo supplementation (HP). Since amino acids are the building blocks of proteins, we included the HP diet group to examine if the supplementation of standard-protein diet with BCAA could attain the same outcomes as that of a high-protein diet. The BCAA diet is the experimental diet intervention, while the CT and HP diets are placebo diets that served as controls for assessing the effect of BCAA supplementation on the study outcomes. The three diets were designed to induce an energy deficit of 500 kcal per day. All subjects underwent 16 weeks of dietary weight loss intervention with prescribed meals (Charoen Pokphand Intertrade (PTE) Ltd., Singapore, JR Foods (PTE) Ltd., Singapore) and daily supplements (BCAA or placebo), which were provided according to body weight (BCAA: 0.1 g·kg^−1^ body weight·d^−1^ divided into three doses per day with main meals, maximum 11.05 g/day). The proposed dose of BCAA at 0.1 g·kg^−1^ body weight·d^−1^ was based on the typical dose used in other studies [30,41,42], and the maximum amount given was below the upper tolerable limit of BCAA [43]. Weekly food delivery for 6 days’ worth of food was provided throughout the 16 weeks, and subjects had one ‘free day’ each week of their own food. The placebo and BCAA supplements were provided throughout the study. The BCAA (AST BCAA 4500, Golden, CO, USA) had 325 mg L-leucine, 162.5 mg L-isoleucine, and 162.5 mg L-valine per tablet with no other added vitamins, minerals, and nutrients. Placebo was manufactured (Beacons Pharmaceuticals Pte Ltd., Singapore) to resemble the BCAA in appearance but consisted of corn-starch and magnesium stearate.

To document diet compliance, subjects checked off food checklists sent with their weekly food delivery and recorded extra foods and beverages consumed. The dietitian provided diet counselling sessions at weeks 0, 4, 8, 12, and 16, and also called subjects fortnightly to assess and encourage compliance. 

The subjects included in this study attended 2 study visits: baseline (week 0) and after controlled dietary intervention (week 16). At each study visit (week 0 and week 16), the subjects consumed a 600 kcal test meal representative of their study diet within 20 min. The test meal was in the form of a packaged meal and a drink. The standard-protein test meal consisted of 14% protein, 56% carbohydrate, and 30% fat, and the amount of BCAA additionally provided to the BCCA group was approximately half the dose of a day’s allocated intake. The high-protein test meal consisted of 27% protein, 43% carbohydrate, and 30% fat.

### 2.3. Anthropometric and Biochemical Measurements 

Anthropometric measurements including height, weight, waist circumference, and blood pressure were recorded [34]. Height and weight were taken using a Seca 763 Digital scale. Blood pressure was taken after subjects rested for at least 20 min using an Omron Digital Automatic Blood Pressure Monitor (Model HEM-907). Body composition (percent body fat, fat mass, and lean mass) was measured using a Dual-energy X-ray Absorptiometry System (DXA) (Hologic, Santa Clara, CA, USA, model: ASY-05119). 

Fasting blood samples were obtained before consumption of test meal during the 2 study visits: week 0 and week 16 for the measurement of plasma glucose (photometric assay, hexokinase method), insulin (immunoenzymatic assay), and lipids (enzymatic colorimetric assay) levels [34]. Homeostatic model assessment of insulin resistance (HOMA-IR) was calculated as previously described [44]. Blood samples were also collected at 180 min after test meal for the measurement of plasma glucose and insulin levels. 

Subjects were asked to maintain their baseline physical activity level throughout the study. All subjects were provided with an accelerometer (ActiGraph, Pensacola, FL, USA) at week 0 and 8 visits, and the accelerometer was worn for a week at both visits. The data were used to calculate metabolic equivalent of task (MET) and percentage of moderate to vigorous-intensity physical activity (%MVPA). There were no significant differences for MET or %MVPA among subjects in the different diet groups [34].

### 2.4. Indirect Calorimetry

RMR and substrate utilization were assessed using an open-circuit indirect calorimetry system that measures the amount of oxygen consumption (VO2) and amount of carbon dioxide production (VCO2). After an 8 h overnight fast, subjects laid supine on a bed in a thermoneutral environment with a clear plastic hood over their head and shoulders, and VO2 and VCO2 were recorded for 30 min (Quark CPET, COSMED, Rome, Italy). The first 10 min of data were discarded to ensure all subjects had reached equilibrium, and the remaining 20 min of data were averaged to calculate resting metabolic rate (RMR), respiratory quotient (RQ), percentage of fat oxidation, and percentage of carbohydrate oxidation [45,46]. RMR is calculated using the Weir formula [47], and RQ is calculated as VCO2/VO2. RQ provides an estimate for transition between percentage of fat (RQ = 0.7) and carbohydrate oxidation (RQ = 1.0) [48]. After the measurement at fasting state, each subject consumed a 600 kcal test meal representative of their study diet within 20 min, and VO2 and VCO2 were recorded continuously for a subsequent 3 h. 

### 2.5. Statistical Analysis 

All analyses were performed using SPSS 26.0 with level of statistical significance set at two-sided *p* < 0.05. Descriptive statistics for numerical and categorical variables are presented as mean ± standard deviation (mean ± SD) and N, respectively. Differences in clinical characteristics among the three diet groups were analyzed by one-way ANOVA with Bonferroni correction for continuous parameters and chi-squared for categorical parameters. Paired *t*-test was used to analyze differences in clinical characteristics before (week 0) and after (week 16) dietary intervention. Spearman’s correlation was used to correlate change in RMR and substrate utilization (week 16 vs. week 0) with change in metabolic parameters (week 16 vs. week 0). Repeated measures ANOVA (within subject group) was used to compare postprandial RQ, fat oxidation, carbohydrate oxidation, and DIT responses after test meal between week 0 and week 16, and between subjects in the three diet groups.

## 3. Results

### 3.1. Clinical Characteristics of Study Participants

There were no significant differences in the baseline (week 0) characteristics for gender, age, weight, BMI, waist circumference, percent body fat, fat and lean mass, blood pressure, RMR, fat oxidation, carbohydrate oxidation, RQ, lipid profile and parameters of glucose homeostasis among the three diet groups (Appendix A). 

After 16 weeks of weight loss intervention, the study participants showed significant reductions in weight, BMI, waist circumference, percent body fat, total body fat mass, total body lean mass, systolic blood pressure, diastolic blood pressure, RMR, total cholesterol, triglycerides, HDL cholesterol, LDL cholesterol, fasting glucose, fasting insulin, and HOMA-IR (Appendix A). 

However, there were no significant differences in the absolute change in RMR, RQ, and metabolic parameters among the three diet groups after 16 weeks of weight loss intervention (Table 1).

### 3.2. Correlation between Change in RMR and Change in Metabolic Measures after 16 Weeks of Weight Loss Intervention (Week 16 vs. Week 0)

The change in RMR was positively correlated with the change in weight, BMI, and total body fat mass (Figure 1). A decrease in RMR was correlated with a decrease in weight (r = 0.225, *p* = 0.017) (Figure 1A), BMI (r = 0.206, *p* = 0.030) (Figure 1B), and total body fat mass (r = 0.193, *p* = 0.042) (Figure 1C). The change in RMR was not significantly correlated with the change in metabolic parameters within each diet group (data not shown).

### 3.3. Correlation between Change in RQ and Change in Metabolic Measures after 16 Weeks of Weight Loss Intervention (Week 16 vs. Week 0)

The change in RQ was correlated with the change in HDL cholesterol (r = 0.219, *p* = 0.021), fasting insulin (r = 0.224, *p* = 0.018), HOMA-IR (r = 0.226, *p* = 0.017), and systolic blood pressure (r = 0.217, *p* = 0.022) (Figure 2A–D). Within groups, the change in RQ was not correlated with the change in metabolic parameters in the CT and BCAA groups (data not shown). However, the change in RQ was significantly correlated with the change in HDL cholesterol (r = 0.355, *p* = 0.037) and percent body fat (r = −0.329, *p* = 0.041) in the HP group.

### 3.4. Postprandial RQ, Fat Oxidation, Carbohydrate Oxidation, and DIT Responses before and after 16 Weeks of Weight Loss Intervention

Overall, there were no significant differences in mean postprandial responses for fat oxidation, carbohydrate oxidation, and RQ before and after weight loss intervention (Figure 3A–C), except that the mean postprandial DIT response was significantly lower (*p* < 0.0005) after weight loss intervention (Figure 3D).

### 3.5. Postprandial RQ, Fat Oxidation, Carbohydrate Oxidation, and DIT Responses among the Three Diet Groups at Week 16 of Diet Intervention

At baseline (week 0), there were no significant differences in mean postprandial responses for RQ, fat oxidation, carbohydrate oxidation, and DIT among the three diet groups (Appendix A).

After 16 weeks of weight loss intervention, the BCAA diet group had significantly lower mean postprandial RQ (*p* = 0.028) (Figure 4A) and carbohydrate oxidation (*p* = 0.044) (Figure 4C), but higher mean postprandial fat oxidation response (*p* = 0.021) (Figure 4B) as compared to the CT group. There were no significant differences in mean postprandial DIT response among the three diet groups (*p* = 0.232) (Figure 4D). The postprandial trajectories for RQ, fat oxidation, carbohydrate oxidation, and DIT were significantly different between the three diet groups (all group × time interaction *p* < 0.05) (Figure 4). 

## 4. Discussion

In this randomized single-blinded, placebo-controlled trial, we showed that there were significant overall improvements in anthropometric measurements and metabolic parameters after 16 weeks of hypocaloric diet intervention [34]. We also observed a significant overall reduction in RMR after 16 weeks of weight loss intervention. Our findings are consistent with other studies that demonstrated a decrease in RMR after weight loss [8,9]. However, increasing protein content or supplementing BCAA in the diet did not result in differential effects on anthropometric measurements, metabolic parameters, or RMR. 

While human and animal studies have demonstrated that BCAA supplementation could increase RMR [35,36,37], our data showed that BCAA supplementation was unable to minimize the reduction of RMR after weight loss. The significant decrease in RMR and DIT after weight loss is likely attributed to metabolic adaptation, where the body alters the rate of energy metabolism as a response to weight loss or energy restriction [10,11]. Although lean body mass was reported to be correlated with RMR [5,7], we did not observe a significant correlation between the change in RMR and the change in total body lean mass after weight loss. The decline in RMR is likely due to metabolic adaptation to prevent continuous weight loss and not accounted for by the loss in lean mass [49]. RQ reflects substrate utilization and the change in RQ represents a change in the type of substrate, i.e., carbohydrate and fat that is oxidized for energy metabolism [48]. The utilization of different substrates has been implicated in metabolic complications [50], and we showed that the change in RQ (substrate utilization) was positively correlated with the change in metabolic outcomes such as HDL cholesterol, fasting insulin, HOMA-IR, and systolic blood pressure.

Our data also demonstrated a significant difference in the postprandial RQ, fat oxidation, and carbohydrate oxidation responses between hypocaloric standard-protein diet with and without BCAA supplementation (BCAA vs. CT). BCAA supplementation significantly lowered the postprandial RQ, indicating a higher fat but lower carbohydrate oxidation response, suggesting that BCAA-supplemented weight loss intervention might promote postprandial fat oxidation response. Further studies are required to examine the long-term effect of the BCAA-induced postprandial fat oxidation response on body fat composition. BCAA supplementation has been found to increase lipid oxidation during exercise in glycogen-depleted subjects [39]. Mourier et al. showed that a BCAA-supplemented hypocaloric diet produced the highest body fat loss in male wrestlers with a high level of exercise performance as compared to hypocaloric diets without BCAA supplementation, and it was postulated that the fat loss may be caused by specific hormonal adaptations induced by BCAA [32]. Dudgeon et al. also showed that consumption of a BCAA-supplemented hypocaloric diet during resistance training caused a greater decrease in body fat percentage as compared to hypocaloric diet without BCAA supplementation [31]. The differences in study cohort, caloric value of diet, dosage of BCAA supplements, duration of study, and exercise intensity between our study and those of Mourier et al. and Dudgeon et al. [31,32] may explain the absence of significant body fat loss between BCAA and CT groups in our study. The decreased rate of carbohydrate oxidation within the BCAA group after 16 weeks of weight loss intervention may indicate the preferential ability of BCAA-supplemented diet in oxidizing fat instead of carbohydrate. L-Leucine activates mTOR signaling pathway to stimulate protein synthesis in the skeletal muscle and promote mitochondrial biogenesis, which results in increased cellular respiration. The increase in cellular energy metabolism stimulates fatty acid oxidation [51]. Hence, BCAA supplementation to a diet that is low in carbohydrate and high in fat may promote greater body fat loss [52].

The strength of our study lies in it being a randomized controlled trial, which included the supply of meals and monitoring of food intake, and BCAA supplementation was provided for a longer period of 16 weeks to a larger cohort of subjects (*n* = 35) as compared to previous studies [53]. The main limitation in this study is that we were unable to calculate protein oxidation as we did not measure urinary nitrogen excretion. However, the aim of our study was to examine the effect of diets of different protein content on RMR and substrate utilization in terms of RQ, and thus the information on protein oxidation would not add to the results of this study.

## 5. Conclusions

In conclusion, BCAA supplementation to a standard-protein hypocaloric diet did not significantly minimize the decrease in RMR compared to standard-protein and high-protein hypocaloric diets without BCAA supplementation. However, the postprandial fat oxidation response was higher after BCAA-supplemented weight loss intervention. 

## Figures and Tables

**Figure 1 nutrients-13-04245-f001:**
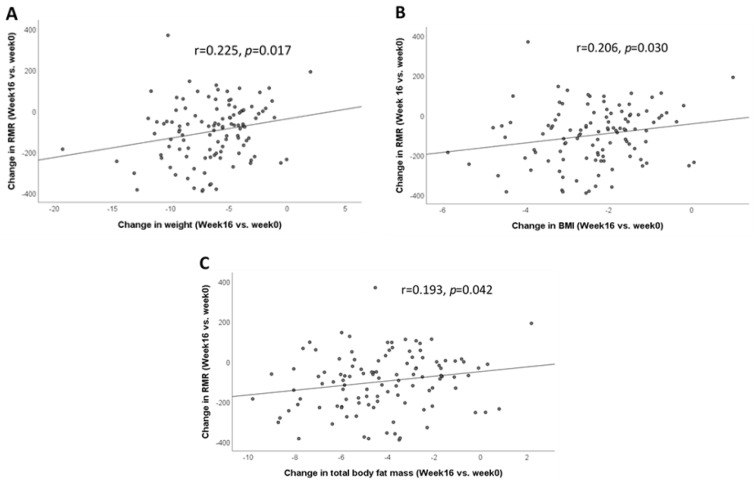
Correlation between change in RMR and change in anthropometric measures after 16 weeks of dietary weight loss intervention (week 16 vs. week 0). (**A**) Correlation between the change in RMR and the change in weight; (**B**) correlation between the change in RMR and the change in BMI; (**C**) correlation between the change in RMR and the change in total body fat mass. Data were analyzed by Spearman’s correlation. Abbreviations: body mass index, BMI; resting metabolic rate, RMR.

**Figure 2 nutrients-13-04245-f002:**
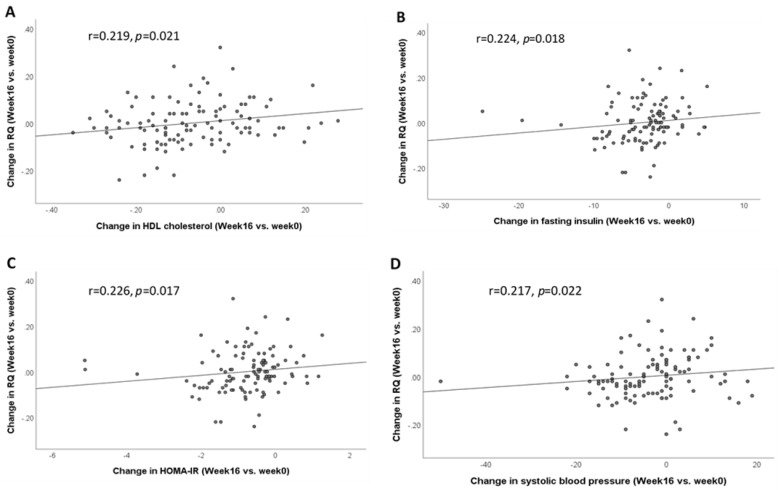
Correlation between the change in RQ and the change in metabolic measures after 16 weeks of dietary weight loss intervention (week 16 vs. week 0). (**A**) Correlation between the change in RQ and the change in HDL cholesterol; (**B**) correlation between the change in RQ and the change in fasting insulin; (**C**) correlation between the change in RQ and the change in HOMA-IR; (**D**) correlation between the change in RQ and the change in systolic blood pressure. Data were analyzed by Spearman’s correlation. Abbreviations: high-density lipoprotein, HDL; homeostatic model assessment of insulin resistance, HOMA-IR; respiratory quotient, RQ.

**Figure 3 nutrients-13-04245-f003:**
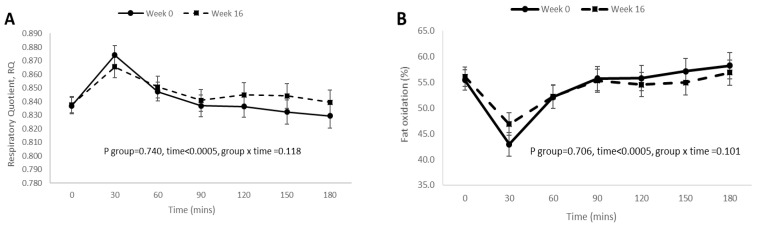
Postprandial RQ, fat oxidation, carbohydrate oxidation, and DIT responses after 16 weeks of weight loss intervention. (**A**) Postprandial RQ responses after 16 weeks of weight loss intervention; (**B**) postprandial fat oxidation responses after 16 weeks of weight loss intervention; (**C**) postprandial carbohydrate oxidation responses after 16 weeks of weight loss intervention; (**D**) postprandial DIT responses after 16 weeks of weight loss intervention. Data were presented as mean ± standard error at each time-point and analyzed by repeated measures ANOVA (within subject group).

**Figure 4 nutrients-13-04245-f004:**
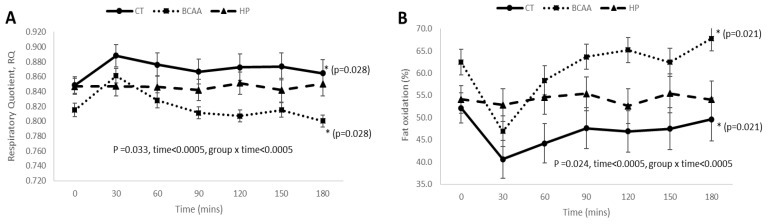
Postprandial RQ, fat oxidation, carbohydrate oxidation, and DIT responses among the three diet groups after 16 weeks of weight loss intervention. (**A**) Postprandial RQ responses among the 3 diet groups; (**B**) postprandial fat oxidation responses among the 3 diet groups; (**C**) postprandial carbohydrate oxidation responses among the 3 diet groups; (**D**) postprandial DIT responses among the 3 diet groups. Data were presented as mean ± standard error and analyzed by repeated measures ANOVA (between subject groups). Abbreviations: standard-protein with BCAA, BCAA; standard-protein with placebo, CT; high-protein with placebo, HP. * denotes significant difference of *p* < 0.05 between groups.

**Table 1 nutrients-13-04245-t001:** Change in clinical parameters among study participants in the three diet groups after 16 weeks of weight loss intervention.

Parameters	CT (*n* = 37)	BCAA (*n* = 35)	HP (*n* = 39)	*p*
Weight (kg)	−6.62 ± 3.44	−6.39 ± 3.99	−6.35 ± 2.37	0.931
BMI (kg/m^2^)	−2.45 ± 1.36	−2.26 ± 1.33	−2.33 ± 0.811	0.780
Waist circumference (cm)	−6.32 ± 3.69	−7.43 ± 4.55	−7.18 ± 3.12	0.427
Body fat percentage (%)	−2.34 ± 1.95	−2.37 ± 2.41	−2.75 ± 1.92	0.631
Total body fat mass (kg)	−4.23 ± 2.24	−4.09 ± 2.88	−4.34 ± 1.91	0.902
Total body lean mass (kg)	−2.39 ± 1.50	−2.15 ± 1.74	−1.80 ± 1.52	0.272
RMR (kcal/day)	−98.6 ± 165	−108 ± 119	−90.8 ± 140	0.874
Fat (%)	−2.35 ± 27.0	4.06 ± 23.4	−0.95 ± 32.6	0.599
CHO (%)	2.35 ± 26.9	−4.09 ± 23.4	0.95 ± 32.5	0.595
RQ	0.008 ± 0.09	−0.012 ± 0.07	0.006 ± 0.11	0.599
Total cholesterol (mmol/L)	−0.32 ± 0.60	−0.25 ± 0.51	−0.36 ± 0.48	0.648
Triglycerides (mmol/L)	−0.13 ± 0.38	−0.17 ± 0.54	−0.23 ± 0.31	0.581
HDL cholesterol (mmol/L)	−0.08 ± 0.13	−0.07 ± 0.14	−0.05 ± 0.12	0.469
LDL cholesterol (mmol/L)	−0.17 ± 0.47	−0.10 ± 0.49	−0.21 ± 0.40	0.585
Fasting glucose (mmol/L)	−0.19 ± 0.32	−0.15 ± 0.25	−0.16 ± 0.31	0.874
Fasting insulin (mU/L)	−3.54 ± 4.21	−3.60 ± 4.96	−2.96 ± 3.77	0.779
HOMA-IR	−0.84 ± 1.06	−0.79 ± 1.03	−0.68 ± 0.91	0.768
Systolic blood pressure (mmHg)	−3 ± 8	−4 ± 9	−3 ± 12	0.932
Diastolic blood pressure (mmHg)	−3 ± 9	−6 ± 6	−2 ± 11	0.138

Data presented as mean change ± SD. One-way ANOVA with Bonferroni correction was used to compare differences in absolute change in clinical parameters (week 16 vs. week 0) among the 3 diet groups. Abbreviations: standard-protein with BCAA, BCAA; standard-protein with placebo, CT; carbohydrates, CHO; high-density lipoprotein, HDL; high-protein with placebo, HP; homeostatic model assessment of insulin resistance, HOMA-IR; low-density lipoprotein, LDL; resting metabolic rate, RMR; respiratory quotient, RQ.

## Data Availability

The data presented in this study are available within the article or Appendix A.

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
