# Peer review of "Branched Chain Amino Acid Supplementation to a Hypocaloric Diet Does Not Affect Resting Metabolic Rate but Increases Postprandial Fat Oxidation Response in Overweight and Obese Adults after Weight Loss Intervention"

_nutrients, 2021, doi:10.3390/nu13124245_

Round 1
Reviewer 1 Report
Title: Branched chain amino acid supplementation to a hypocaloric diet increases postprandial fat oxidation response in over- 3 weight and obese adults after weight loss intervention
Overall- this paper is an analysis of a RCT that tested if supplementation with BCAA offset RMR decrease after a weight loss intervention compared to standard protein supplementation and high-protein supplementation. The intervention had no impact on RMR but did result in higher fat oxidation. Overall, the paper is very interesting, however a better organization and transparency on what were the primary and secondary outcomes of the original trial and how the comparison groups were set up for analysis would strengthen the paper and facilitate the interpretation of the results.
Abstract – In the first sentence of the abstract – should it say BCAA supplementation is reported to aid…. ? Otherwise please clarify what of BCAA aids in lean mass preservation.
Line 33- please clarify which RMR is before the intervention, and which one is after. Was this reduction significant?
Clarify how fat oxidation was assessed.
Introduction:
All the information provided is very valuable and well-written, however, given that BCAA amino acid supplementation for weight loss is the focus of the paper, I recommend centering the introduction around their importance, their potential impact on weight loss interventions and the evidence behind it. Right now, it is unclear if the focus of the study was on protein supplementation and RMR, and BCAA and fat oxidation was a secondary finding.
Methods
Again, it would be good to clarify which are the trial primary and secondary outcomes. According to the trial registry fat oxidation is not an outcome but RMR is a secondary outcome. It is fine to analyze the trial design for other analyses but this should be very clear in the methods.
Study design
It would be good to clarify that the two arms besides BCAA were considered placebos and that the experimental intervention was BCAA.
Discussion
287 – My understanding was that there was no impact of the intervention (BCAA) on RMR (even though all groups saw a reduction on RMR). Maybe separate from the sentence describing the results on the primary outcomes and clarify that there was an overall decrease in RMR.
327-331 – other limitations/strengths such as attrition, follow-up time, RCT design, study population should be discussed.
I think the title should be modified to reflect the null finding on RMR as well as the additional findings on fat oxidation.
Author Response
We thank the reviewer for the constructive comments and suggestions. We have responded to the comments and made the necessary revisions in the manuscript. All amendments are tracked in the revised manuscript.
Title: Branched chain amino acid supplementation to a hypocaloric diet increases postprandial fat oxidation response in over- 3 weight and obese adults after weight loss intervention
Overall- this paper is an analysis of a RCT that tested if supplementation with BCAA offset RMR decrease after a weight loss intervention compared to standard protein supplementation and high-protein supplementation. The intervention had no impact on RMR but did result in higher fat oxidation. Overall, the paper is very interesting, however a better organization and transparency on what were the primary and secondary outcomes of the original trial and how the comparison groups were set up for analysis would strengthen the paper and facilitate the interpretation of the results.
Abstract – In the first sentence of the abstract – should it say BCAA supplementation is reported to aid…. ? Otherwise please clarify what of BCAA aids in lean mass preservation.
Reply to reviewer: We have now revised the first sentence of the abstract (Page 1, line 26) to “Branched chain amino acids (BCAA) supplementation is reported to aid in” as suggested by the reviewer.
Line 33- please clarify which RMR is before the intervention, and which one is after. Was this reduction significant?
Reply to reviewer: We have now revised the sentence (Page 1, lines 34-35) to “RMR was reduced from 1600±270kcal/day to 1500±264kcal/day (p<0.0005) after weight loss”.
Clarify how fat oxidation was assessed.
Reply to reviewer: We have now revised the sentence (Page 1, lines 32-34) to “Indirect calorimetry was used to measure RMR, carbohydrate and fat oxidation before and after 16 weeks of dietary intervention” to clarify that fat oxidation was assessed by indirect calorimetry.
Introduction:
All the information provided is very valuable and well-written, however, given that BCAA amino acid supplementation for weight loss is the focus of the paper, I recommend centering the introduction around their importance, their potential impact on weight loss interventions and the evidence behind it. Right now, it is unclear if the focus of the study was on protein supplementation and RMR, and BCAA and fat oxidation was a secondary finding.
Reply to reviewer: We thank the reviewer for the comments. We have now revised the introduction section and included more information regarding the impact of BCAA on weight loss, RMR and substrate utilization.
Page 2, lines 80-94 “BCAA supplementation to a moderate protein hypocaloric diet was found to cause the highest body weight loss and decrease in percent body fat as well as a significant reduction in abdominal visceral adipose tissue in a group of competitive wrestlers [32]. However, some studies did not observe an effect of BCAA supplementation on body weight, lean and fat mass in overweight and obese adults after a hypocaloric diet-induced weight loss [33,34].
Preliminary studies showed that BCAA supplementation increased RMR in human adults [35,36]. The knockdown of branched-chain aminotransferase (BCAT) gene which encodes the enzyme involved in BCAA catabolism, resulted in increased plasma BCAAs, decreased adiposity and body weight, and increased energy expenditure in mice [37]. In addition, dietary BCAA restriction has been found to alter substrate utilization in Zucker fatty rats [38], and BCAA supplementation was shown to enhance lipid oxidation during exercise [39]. However, the effect of BCAA supplementation on RMR and substrate utilization during hypocaloric diet-induced weight loss intervention in overweight and obese adults has not been investigated.”
Methods
Again, it would be good to clarify which are the trial primary and secondary outcomes. According to the trial registry fat oxidation is not an outcome but RMR is a secondary outcome. It is fine to analyze the trial design for other analyses but this should be very clear in the methods.
Reply to reviewer: We thank the reviewer for the comments. We have now revised the sentences (Page 3, lines 102-107) to “The study participants were part of a randomized, controlled trial registered at clin-cialtrials.gov as NCT02277275. The trial aimed to examine the effect of BCAA supplementation on changes in primary outcomes including lean and fat mass, and changes in secondary outcomes including insulin sensitivity, RMR and diet-induced thermogenesis (DIT)” for clarification on the primary and secondary outcomes of the trial.
Study design
It would be good to clarify that the two arms besides BCAA were considered placebos and that the experimental intervention was BCAA.
Reply to reviewer: We have now included the sentence (Page 3, lines 130-132) “The BCAA diet is the experimental diet intervention, while the CT and HP diets are placebo diets that served as controls for assessing the effect of BCAA supplementation on the study outcomes”.
Discussion
287 – My understanding was that there was no impact of the intervention (BCAA) on RMR (even though all groups saw a reduction on RMR). Maybe separate from the sentence describing the results on the primary outcomes and clarify that there was an overall decrease in RMR.
Reply to reviewer: We have now revised the sentence (Page 9, lines 328-329) to “We also observed a significant overall reduction in RMR after 16 weeks of hypocaloric diet intervention”.
327-331 – other limitations/strengths such as attrition, follow-up time, RCT design, study population should be discussed.
Reply to reviewer: We thank the reviewer for the comment. We have now included more discussion on the strength and limitation of our study.
Page 10, lines 375-382 “The strength of our study lies in it being a randomized controlled trial which included the supply of meals and monitoring of food intake, and BCAA supplementation was provided for a longer period of 16 weeks to a bigger cohort of subjects (n=35) as compared to previous studies [51]. The main limitation in this study is that we were unable to calculate protein oxidation as we did not measure urinary nitrogen excretion. However, the aim of our study was to examine the effect of diets of different protein content on RMR and substrate utilization in terms of RQ, thus the information on protein oxidation would not add to the results of this study.”
I think the title should be modified to reflect the null finding on RMR as well as the additional findings on fat oxidation.
Reply to reviewer: We have now revised the title to “Branched chain amino acid supplementation to a hypocaloric diet does not affect resting metabolic rate but increases post-prandial fat oxidation response in overweight and obese adults after weight loss intervention” as suggested by the reviewer.

Reviewer 2 Report
General comments:
The objective of study submitted by Ooi et al. was to assess the effect of supplemental BCAA in hypocaloric diet on RMR and substrate utilization during a weight loss intervention. Although understanding the effect of BCAA on RMR is important for the field, the idea is not completely novel. The Introduction and Discussion of the paper are missing important literature on the effect of BCAA on energy expenditure and RMR (animal models and clinical trials). The methods need to be enriched by adding further clarification on diets and analysis as highlighted in my specific comments. In results section, the graphs can be improved by adding more details on analysis and variations.
Specific comments:
Line 94: Assuming differences in RMR in people with different ages, did authors separate the effect of age in their analysis?
Line 103: although the effect of gender and BMI are blocked, from the results it is not clear whether the effect of block was significant or insignificant
Line 109: Please provide the diets in a separate Table
Line 112-114: please provide a justification for the used doses
Line 127: please provide more details on dietary composition of the “test meal” either in a chart or in the text
Line 159: how 20 min of data collected for RMR can be extrapolated for total RMR of a subject?
Line 256: please add the error bars and the significance symbols for each time point for Fig. 1
Line 276: same comment as Fig. 1
Line 286: according to Table 1, there are no significant improvements in anthropometric measurements
Line 333-335: please rephrase as it is not clear
Author Response
We thank the reviewer for the constructive comments and suggestions. We have responded to the comments and made the necessary revisions in the manuscript. All amendments are tracked in the revised manuscript.
General comments:
The objective of study submitted by Ooi et al. was to assess the effect of supplemental BCAA in hypocaloric diet on RMR and substrate utilization during a weight loss intervention. Although understanding the effect of BCAA on RMR is important for the field, the idea is not completely novel. The Introduction and Discussion of the paper are missing important literature on the effect of BCAA on energy expenditure and RMR (animal models and clinical trials). The methods need to be enriched by adding further clarification on diets and analysis as highlighted in my specific comments. In results section, the graphs can be improved by adding more details on analysis and variations.
Reply to the reviewer: We thank the reviewer for the comments. We have now included more literature in the introduction and discussion, and we have responded to the reviewer’s specific comments.
Introduction:
Page 2, lines 80-94 “BCAA supplementation to a moderate protein hypocaloric diet was found to cause the highest body weight loss and decrease in percent body fat as well as a significant reduction in abdominal visceral adipose tissue in a group of competitive wrestlers [32]. However, some studies did not observe an effect of BCAA supplementation on body weight, lean and fat mass in overweight and obese adults after a hypocaloric diet-induced weight loss [33,34].
Preliminary studies showed that BCAA supplementation increased RMR in human adults [35,36]. The knockdown of branched-chain aminotransferase (BCAT) gene which encodes the enzyme involved in BCAA catabolism, resulted in increased plasma BCAAs, decreased adiposity and body weight, and increased energy expenditure in mice [37]. In addition, dietary BCAA restriction has been found to alter substrate utilization in Zucker fatty rats [38], and BCAA supplementation was shown to enhance lipid oxidation during exercise [39]. However, the effect of BCAA supplementation on RMR and substrate utilization during hypocaloric diet-induced weight loss intervention in overweight and obese adults has not been investigated.”
Discussion:
Page 10, lines 334-336 “While human and animal studies have demonstrated that BCAA supplementation could increase RMR [35-37], our data showed that BCAA supplementation is unable to minimize the reduction of RMR after weight loss.”
Page 10, lines 356-357 “BCAA supplementation has been found to increase lipid oxidation during exercise in glycogen-depleted subjects [39].”
Specific comments:
Line 94: Assuming differences in RMR in people with different ages, did authors separate the effect of age in their analysis?
Reply to the reviewer: We thank the reviewer for the comment. We did not adjust/separate the effect of age in our analyses because there were no significant differences in the age of the subjects between the 3 diet groups as shown in supplementary Table S1.
Line 103: although the effect of gender and BMI are blocked, from the results it is not clear whether the effect of block was significant or insignificant
Reply to the reviewer: We thank the reviewer for the comment. After the subjects were block randomized by gender and BMI groups into the 3 hypocaloric diet groups, there were no significant differences in the gender distribution and BMI between the 3 diet groups as shown in supplementary Table S1.
Line 109: Please provide the diets in a separate Table
Reply to the reviewer: We thank the reviewer for the comment. The dietary composition for each diet group (CT, BCAA and HP) and examples of the food provided to the participants are available in our previously published report. We have referenced our published report [34] in page 3, line 123.
Line 112-114: please provide a justification for the used doses
Reply to the reviewer: We have now provided justification for the used dose of BCAA in page 3, lines 137-139 “The proposed dose of BCAA at 0.1g ∙ kg-1 body weight ∙ d-1 was based on the typical dose used in other studies [30,41,42], and the maximum amount given was below the upper tolerable limit of BCAA [43].”
Line 127: please provide more details on dietary composition of the “test meal” either in a chart or in the text
Reply to the reviewer: We thank the reviewer for the comment. We have provided the dietary composition of the test meal for the 3 diet groups (CT-standard protein, BCAA-standard protein with BCAA, HP-high protein) in the text (page 4, lines 152-156).
Line 159: how 20 min of data collected for RMR can be extrapolated for total RMR of a subject?
Reply to the reviewer: We thank the reviewer for the comment. Based on the operating instruction and manual of the indirect calorimetry system (Quark CPET, COSMED), 20mins of data were averaged to obtain the average amount of oxygen consumption (VO2) and amount of carbon dioxide production (VCO2) of a subject at resting state. RMR is calculated using the Weir formula with the average VO2 and VCO2 values. We have stated that the 20mins of data were averaged to calculate RMR (Page 4, line 185).
Line 256: please add the error bars and the significance symbols for each time point for Fig. 1
Reply to the reviewer: We thank the reviewer for the comment. Since Figures 1 and 2 are scatterplots for correlation analyses, we believed the reviewer is referring to Figure 3 (line 256 as highlighted by the reviewer-current line 283). We have decided not to report the standard deviations using error bars as the error bars overlapped significantly between the 3 different diet groups, making the figure less meaningful for interpretation. The change over time between groups for each graph was not significant (P value for analysis between groups (P group) was included in the graph). Hence, there were no significance symbols in the graph.
Example of the figure with error bar is provided.
Line 276: same comment as Fig. 1
Reply to the reviewer: We thank the reviewer for the comment. We have decided not to report the standard deviations using error bars as the error bars overlapped significantly between the 3 different diet groups, making the figure less meaningful for interpretation.
Example of the figure with error bar is provided.
Line 286: according to Table 1, there are no significant improvements in anthropometric measurements
Reply to the reviewer: We thank the reviewer for the comment and apologized for the confusion. Table 1 shows the comparison of the change in clinical parameters among the 3 diet groups after dietary intervention. However, in lines 326-329 (page 9) of the Discussion section, we were trying to explain that there were significant overall improvements in clinical parameters and RMR after 16 weeks of dietary intervention regardless of diet group (Supplementary Table S2).
We have now revised the sentences (Page 9, lines 326-329) to “In this randomized single-blinded, placebo-controlled trial, we showed that there were significant overall improvements in anthropometric measurements and metabolic parameters after 16 weeks of hypocaloric diet intervention [32]. We also observed a significant overall reduction in RMR after 16 weeks of weight loss intervention” for clarifications.
Line 333-335: please rephrase as it is not clear
Reply to the reviewer: We thank the reviewer for the comment. We have now revised the sentences (Page 11, lines 388-392) to “In conclusion, BCAA supplementation to a standard-protein hypocaloric diet did not significantly minimize the decrease in RMR compared to standard-protein and high-protein hypocaloric diets without BCAA supplementation. However, the postprandial fat oxidation response increases after BCAA-supplemented weight loss intervention”.

Round 2
Reviewer 2 Report
The authors have addressed the majority of my concerns and have applied the necessary changes in the revised manuscript. However, I’m still not convinced about the authors reply to 2 of my previous comment:
- “how 20 min of data collected for RMR can be extrapolated for total RMR of a subject”?
I understand that authors have averaged the 20 min data to calculate RMR. Extrapolating the results of 20 min measurement to whole day RER require caution while the subjects go through of several fast or feast cycles during the day that can influence the RER. Following simply the instruction of producer does not justify the way calculations have been dome. Authors either should support their calculation based on previous literature or state that as the limitation of the study.
- “please add the error bars and the significance symbols for each time point for Fig. 1”
Sorry, yes I was referring to Fig. 3. According to authors the error bars are overlapping among 3 diets and changes over time between groups for each graph was not significant. Then, I disagree with the authors that adding error bars will be “making the figure less meaningful for interpretation” as the way they are shown now are misleading and give the impression that the groups are different, but statically they are not. Authors may decide to show only 1 side of error bars.
Author Response
We thank the reviewer for the constructive comments and suggestions. We have responded to the comments and made the necessary revisions in the manuscript. All amendments are tracked in the revised manuscript.
The authors have addressed the majority of my concerns and have applied the necessary changes in the revised manuscript. However, I’m still not convinced about the authors reply to 2 of my previous comment:
“how 20 min of data collected for RMR can be extrapolated for total RMR of a subject”?
I understand that authors have averaged the 20 min data to calculate RMR. Extrapolating the results of 20 min measurement to whole day RER require caution while the subjects go through of several fast or feast cycles during the day that can influence the RER. Following simply the instruction of producer does not justify the way calculations have been dome. Authors either should support their calculation based on previous literature or state that as the limitation of the study.
Reply to reviewer: We thank the reviewer for the comments. We have now cited previous literature (references 45 and 46) to support our calculation.
Reference 45: Luscombe ND, Tsopelas C, Bellon M, Clifton PM, Kirkwood I, Wittert GA (2006) Use of [14C]-sodium bicarbonate/urea to measure total energy expenditure in overweight men and women before and after low calorie diet induced weight loss. Asia Pac J Clin Nutr 15 (3):307-316
Reference 46: Henry CJ, Lightowler HJ, Marchini J (2003) Intra-individual variation in resting metabolic rate during the menstrual cycle. Br J Nutr 89 (6):811-817. doi:10.1079/BJN2003839
“please add the error bars and the significance symbols for each time point for Fig. 1”
Sorry, yes I was referring to Fig. 3. According to authors the error bars are overlapping among 3 diets and changes over time between groups for each graph was not significant. Then, I disagree with the authors that adding error bars will be “making the figure less meaningful for interpretation” as the way they are shown now are misleading and give the impression that the groups are different, but statically they are not. Authors may decide to show only 1 side of error bars.
Reply to reviewer: We thank the reviewer for the comment. We have now revised Figures 3 and 4 to include the standard error bars in the graphs.
